# High-Sensitivity Troponin T: A Potential Safety Predictive Biomarker for Discharge from the Emergency Department of Patients with Confirmed Influenza

**DOI:** 10.3390/jpm12040520

**Published:** 2022-03-23

**Authors:** Manuel Antonio Tazón-Varela, Jon Ortiz de Salido-Menchaca, Pedro Muñoz-Cacho, Enara Iriondo-Bernabeu, María Josefa Martos-Almagro, Emma Lavín-López, Ander Vega-Zubiaur, Edgar José Escalona-Canal, Iratxe Alcalde-Díez, Carmen Gómez-Vildosola, Ainhoa Belzunegui-Gárate, Fabiola Espinoza-Cuba, José Antonio López-Cejuela, Alba García-García, Alejandro Torrejón-Cereceda, Elena Sabina Nisa-Martínez, Diana Moreira Nieto, Cintia Hellín-Mercadal, Ander García-Caballero, Héctor Alonso-Valle

**Affiliations:** 1Servicio de Urgencias, Hospital de Laredo, 39770 Laredo, Spain; tazovare@yahoo.es (M.A.T.-V.); jonortiz1989@gmail.com (J.O.d.S.-M.); enarairiondo1@gmail.com (E.I.-B.); majos1709@hotmail.com (M.J.M.-A.); emmuca@hotmail.com (E.L.-L.); andervz@hotmail.com (A.V.-Z.); edgar.escalona.canal@gmail.com (E.J.E.-C.); iratxealkalde@gmail.com (I.A.-D.); cgomezvildo@hotmail.com (C.G.-V.); ainhobelzu@gmail.com (A.B.-G.); fecuba@hotmail.com (F.E.-C.); joselc7@hotmail.es (J.A.L.-C.); albagg95@hotmail.com (A.G.-G.); alextorcer@gmail.com (A.T.-C.); hnmhelen@gmail.com (E.S.N.-M.); dayana_m_n@yahoo.es (D.M.N.); cintiahellin@gmail.com (C.H.-M.); ander.garcia38@gmail.com (A.G.-C.); 2Grupo Salud Comunitaria, Instituto de Investigación Sanitaria Valdecilla (IDIVAL), 39011 Santander, Spain; 3Unidad Docente Gerencia de Atención Primaria, Servicio Cántabro de Salud, 39011 Santander, Spain; 4Servicio de Urgencias, Hospital Universitario Marqués de Valdecilla, Santander, 39008 Santander, Spain; jefaturaestudios.humv@scsalud.es

**Keywords:** influenza, human, troponin, biomarkers, SARS-CoV-2, cardiovascular infections, virus diseases, usTnT

## Abstract

The purpose of the study was to analyze the relationship between the high-sensitivity troponin T levels in patients with confirmed influenza virus infection and its severity determined by mortality during the care process. In addition, a high-sensitivity troponin T cut-off value was sought to allow us to a safe discharge from the emergency department. An analytical retrospective observational study was designed in which high-sensitivity troponin T is determined as an exposure factor, patients are followed until the resolution of the clinical picture, and the frequency of mortality is analyzed. We included patients ≥ 16 years old with confirmed influenza virus infection and determination of high-sensitivity troponin T. One hundred twenty-eight patients were included (96.9% survivors, 3.1% deceased). Mean and median blood levels of high-sensitivity troponin T of survivors were 26.2 ± 58.3 ng/L and 14.5 ng/L (IQR 16 ng/L), respectively, and were statistically different when compared with those of the deceased patients, 120.5 ± 170.1 ng/L and 40.5 ng/L (IQR 266.5 ng/L), respectively, *p* = 0.012. The Youden index using mortality as the reference method was 0.76, and the cut-off value associated with this index was 24 ng/L (sensitivity 100%, specificity 76%, NPV 100%, PPV 4%) with AUC of 88,8% (95% CI: 79.8–92.2%), *p* < 0.001. We conclude that high-sensitivity troponin T levels in confirmed virus influenza infection are a good predictor of mortality in our population, and this predictor is useful for safely discharging patients from the emergency department.

## 1. Introduction

Influenza virus infection (IVI) is a substantial global public health problem that causes significant morbidity and mortality. Annual epidemics are estimated to infect 5–10% of the world’s population, causing 3–5 million severe cases and more than 650,000 deaths [1,2]. In the 2019–2020 season, the impact of the flu epidemic in Spain caused 27,700 hospitalizations with confirmed influenza infection, 1800 patients in the intensive care unit (ICU) services, and 3900 deaths [3].

Although IVI is a usually self-limited pathology that normally infects only the upper respiratory tract, sometimes in relation to its aggressiveness and the characteristics of the host, it can have a complexity and severity that make it difficult for us to predict the evolution upon arrival at health services. In addition, the appearance of the SARS-CoV-2 virus has produced a significant distortion, with a drastic decrease in the incidence of IVI. In the 2020–2021 season by the date of 2 May 2021, 15 IVIs had been reported in Spain, 2 from sentinel samples and 13 nonsentinel samples [4]. We do not know if in the future the influenza virus will disappear in its fight with other viruses in the same ecological niche or will rebound and be a promoter of other infections, but the European Center for Disease Prevention and Control and the World Health Organization (WHO) propose establishing sentinel surveillance systems for influenza, COVID-19 and any other respiratory virus or emerging etiological agent in the future [5].

A growing body of evidence says that infectious diseases cause cardiovascular complications in the short and medium term. There is increasing evidence of the relationship between bacterial pneumonia or COVID-19 infection and important cardiovascular complications with increased biomarkers [6,7,8]. IVI is not an exception, increasing deaths from cardiovascular problems during influenza epidemics and triggering or aggravating episodes of arrhythmias, acute coronary syndrome, acute myocarditis, or acute heart failure [9]. During the last century, the healthcare community has observed cardiovascular complications in patients with IVI. The first reference to the increase in cardiovascular morbidity and mortality during IVI epidemics was described in 1932 by Dr. Selwyn D. Collins [10].

Several large epidemiological studies in Russia, the United States, the United Kingdom, and Hong Kong have revealed a temporal association between influenza virus circulation and increased deaths from ischemic heart disease [11]. In 2015 a meta-analysis determined that a recent diagnosis of IVI doubled the risk of developing an acute coronary syndrome [12], and this risk was found to be able to increase up to 6 times in the seven days after IVI confirmation [13]. There is increasing evidence that a recent flu infection is associated with an increased risk of developing acute coronary syndrome, heart failure, myocarditis, and arrhythmias, increasing hospital admissions and mortality [12,13,14]. In a study of 600 patients with confirmed IVI, 86% of the events associated with acute cardiac injury occurred during the three days after flu laboratory ratification [15]. In short, IVI can predispose to suffering a coronary event in the following days. IVI can also cause other cardiovascular complications, multiplying 3–5 times the risk of heart failure, acute lung edema, and arrhythmias, especially in the first three days of infection [16].

A recent systematic review that analyzed 14 articles that evaluated the elevation of any type of troponin (conventional T and I; ultrasensitive T and I) in patients with IVI concluded that troponin elevation is a rare phenomenon but that when it occurs it increases the risk of death [10].

Searching for a safe cut-off point would decant the patients who could potentially be discharged from the emergency department, with the potential financial savings for the health system in a disease with a large absolute number of seasonal infections.

Therefore, we propose a study whose objective is to determine the prognostic capacity of hsTnT to detect mortality in patients with confirmed IVI, searching for a cut-off point that optimizes the sensitivity and negative predictive value (NPV) of the diagnostic test.

## 2. Materials and Methods

### 2.1. Enrolled Patients’ Characteristics

An analytical retrospective observational study was designed in a 145-bed hospital in northern Spain, covering a population of 104,800 people. Subjects ≥ 16 years of age with confirmed IVI and determination of usTnT in the emergency room were included and were followed up until resolution of the condition. The patient inclusion period was from 1 January 2009 to 31 December 2020.

Demographic variables; comorbidity; clinical characteristics; and laboratory values including biomarkers, radiological data, and evolution data were collected.

### 2.2. Diagnostic

The outcome predictor variable was the blood determination of hsTnT (measured in ng/L) and mortality during the care process as a dependent variable.

For the diagnosis of IVI, oropharyngeal samples were collected with swabs, which were analyzed by a rapid diagnostic test that uses an immunochromatographic analysis for the qualitative detection of influenza nucleoprotein antigens and/or molecular biological diagnosis by genomic amplification techniques by polymerase chain reaction (RT-PCR) methods. We consider IVI confirmed by the positivity of either of the two methods due to their high specificity.

hsTnT measurement was performed within the first hour of the patient’s arrival at the emergency department. Electrochemiluminescent immunoassay technique was used to determine hsTnT (Roche Elecsys Diagnostics GMBH autoanalyzer, Sandhofer Strasse 116, D-68305 Mannheim, Germany). For the rest of the biomarkers, immunoturbidimetric tests for C-reactive protein and electrochemiluminescent immunoassay for NT-proBNP were used.

### 2.3. Statistical Analysis

For the statistical analysis, the categorical variables were described as absolute value and percentage, and the continuous variables were described by their mean, standard deviations, medians, and interquartile ranges. To assess the differences between the levels of hsTnT and the rest of the quantitative variables in patients with IVI, an analysis was performed between the surviving and nonsurviving groups for each of the variables using the Mann–Whitney U test. To evaluate differences between groups for qualitative variables, the chi-square test or Fisher’s test was used.

For the analysis of hsTnT as a predictor of mortality, the ideal cut-off point was calculated by optimizing the product of sensitivity by specificity, maximizing the Youden index, and using mortality as the reference method, representing the receiver operating characteristic (ROC) curve and area under the curve (AUC).

Values of *p* < 0.05 were considered statistically significant. The analysis was performed with SPSS for Windows, version 25 (IBM Corp. Released 2017. IBM SPSS Statistics for Windows, Version 25.0. Armonk, NY, USA: IBM Corp.), and MedCalc for the diagnostic utility (MedCalc Statistical Software version 19.6 (MedCalc Software bv, Ostend, Belgium; https://www.medcalc.org; accessed on 22 February 2022)

### 2.4. Bioethical Statement

This study was designed following the ethical principles of the Declaration of Helsinki. It was positively evaluated and certified by the Cantabria Clinical Research Ethics Committee (CEIm 2020.082 certificate) and by the Laredo Hospital Teaching Commission.

## 3. Results

During the study period, 489,825 patients were assisted in the emergency department. Of these patients, 1411 patients were diagnosed with flu syndrome. In 920 patients, IVI was confirmed by immunochromatographic analysis and/or genomic amplification techniques. One hundred twenty-eight patients ≥16 years with confirmed IVI and determination of hsTnT were included in the study (Figure 1).

The mean age was 68.8 ± 15.7 years, and 37.1% were women. In the study population, 19.5% had a history of ischemic heart disease, 11.7% had a history of heart failure, 20.3% had chronic obstructive pulmonary disease, and 19.4% had chronic renal failure. The proportion of patients admitted was 67.2%, 65.6% in the conventional ward and 1.6% in the ICU. Of the patients, 3.1% died, with a mean survival of 5.5 days (SD 4.5). The general characteristics of the sample are shown in the Table 1

The mean value of hsTnT in the surviving patients was 26.2 ± 58.3 ng/L, and the median was 14.5 ng/L (IQR 16). In the deceased, the mean was 120.5 ± 170.1 ng/L and median was 40.5 ng/L (IQR 266.5), *p* = 0.012 (Figure 2).

The Youden index was 0.76, and the cut-off point associated with this index was 24 ng/L (sensitivity 100%, specificity 76%, NPV 100%, positive predictive value (PPV) 4%) with area under the ROC curve (AUC) of 88.8% (95% CI: 79.8–92.2%), *p* < 0.001 (Figure 3).

## 4. Discussion

The main complications of IVI are respiratory (primary influenza pneumonia or viral pneumonia superinfected by bacteria), but when it comes to extrapulmonary complications, a systematic review of 218 articles found that acute myocarditis and acute coronary syndrome were among the most frequent clinical entities [11,17].

On the other hand, troponin is a globular protein widely used for the early detection of acute coronary syndrome when its values are higher than the 99th percentile of the normal reference population, but it also increases in situations of heart failure, myocarditis, and arrhythmias [18,19].

At this point, we have to consider the reasons why a clinician requests the determination of a cardiac biomarker in a patient with influenza syndrome. Possibly it is because we assume that the patient is seriously ill and therefore we suspect that the patient is developing silent acute myocardial damage or some complication secondary to infection, such as myocarditis due to direct myocardial injury or due to substances that decrease contractility, myocardial ischemia due to an imbalance between demands and oxygen supply, ischemia due to plaque instability due to transient loss of anticoagulant properties of the endothelium when infiltrated by components of the mononuclear phagocyte system, arrhythmias due to increased sympathetic–adrenal activity, catecholaminergic coronary spasm, etc. We could also conjecture that troponin is requested in polypathological patients due to suspicion of atypical presentation of coronary syndrome or suspicion of pathology where troponin is useful as a prognostic tool, such as in pulmonary thromboembolic disease. Whatever the etiopathogenic mechanism, any of these fatal complications of IVI cause an increase in hsTnT in peripheral blood.

The elevation of troponin in IVI is not described as frequent, but when it occurs, it worsens the prognosis, increasing the risk of death [20].

The body of evidence in this regard is increasing. A study of 1131 patients with IVI found an increase in TnI by 2.9%, with 26.6% of this subgroup dying [21]. A recent retrospective study on 264 patients with IVI confirmed by RT-PCR that stratified patients with IVI according to normality of hsTnT found that hsTnT values were significantly higher in patients who died within 30 days of diagnosis compared to those who survived (*p* < 0.01) [22]. Lippi et al. determined that troponin elevation as an indicator of complicated IVI is a relatively rare phenomenon in patients with IVI, being more likely in elderly patients with significant comorbidities [20]. These results go in the same direction as our study, where we found statistically significant differences in the blood levels of hsTnT between deceased and survivors with IVI (*p* = 0.012).

Therefore, in this context in which acute cardiovascular symptoms are a potential wake-up call to indicate a complication due to severe IVI, hsTnT is postulated as a valuable prognostic tool to detect bad evolution. However, one aspect that has not been studied much is the ability of hsTnT to detect patients with confirmed IVI who can be safely discharged from the emergency department. Pizzini et al., with a mortality of 3.8% and using hsTnT with a cut-off point of 46.4 ng/L, achieved an NPV of 89% to rule out acute cardiac events (acute coronary syndrome, acute heart failure, or arrhythmia) in patients with confirmed IVI [22].

In our sample, whose mortality is 3.1%, using the proposed cut-off point of 24 ng/L to safely discharge patients with confirmed IVI (NPV of 100% with AUC of 88.8% to detect mortality), we would have been able to avoid 52 admissions, that is, 60.5% of them (52/86), reducing the therapeutic effort in the emergency department and the pressure on hospitalization wards. Furthermore, no patient below the cut-off point died or required intensive care unit assistance.

If we take into account the RAE-CMBD (Specialized Health Care Activity Register) data from the annual report of the National Health System, the average cost per admission in Spain is EUR 4,741.94. During the 10 years of the study, 558 patients were admitted (545 in a conventional ward and 13 in an intensive care unit). Therefore, if we apply our model, 338 patients would not have been admitted, with the potential saving in direct costs being EUR 1,602,775.7 [23].

Elevated baseline troponin levels are detected in the elderly, heart disease, diabetics, or patients with chronic renal failure, which may act as a confounder. In our sample, the group of patients who died were significantly older. However, we do not think that these baseline differences invalidate our results, because we did not find statistically significant differences in other comorbidities. Furthermore, the main objective of our study was to detect patients with potentially severe IVI upon arrival at the hospital, regardless of the etiology being that the troponin increased acutely or that the elevation was in a patient who already had chronically high troponin levels. This idea is supported by studies that confirm that hospital mortality in patients with IVI is significantly higher in both acutely and chronically elevated hsTnT patients [24].

The limitations of the study are fundamentally those derived from a retrospective single-center study that requires the clinician’s request for hsTnT as an inclusion criterion, which could lead to a selection bias. Due to the type of study, it was not possible to carry out further determinations of hsTnT or cardiological evaluation. Nor was the time elapsed from the onset of flu-like symptoms to attendance at the emergency department recorded.

We are also aware that age and chronic heart and kidney disease history are potential confounders that may require different cut-off points to avoid interference. Given the small number of ominous events, future studies aimed at consolidating these results will be useful.

Near the end of World War, I an unknown disease ravaged the world’s population, especially the elderly, diabetics, and pregnant and lactating women [25]. The autopsies detected pneumonia and myocarditis. It was the Influenza A virus. We are currently facing another pandemic situation with COVID-19. Both SARS-CoV-2 and influenza are similar in clinical presentation, transmission mechanisms, and vulnerable risk groups and generate similar pulmonary and extrapulmonary complications. In addition, patients with coinfection with IVI and SARS-CoV-2 are more likely to suffer cardiac lesions and earlier cytokine storm [26,27].

The European Center for Disease Prevention and Control and the WHO propose establishing sentinel surveillance systems for influenza and COVID-19 because we do not know what will happen in the near future. A possible scenario would be the coexistence of both viruses, one being a facilitator of the other and both being facilitators of pneumotropic bacteria.

So, in this turbulent period from the microbiological point of view of fighting viruses in health systems, as important as preventive measures with vaccination campaigns aimed at vulnerable groups are, it will also be important to find tools that, once the disease is acquired, allow clinicians to discriminate between stable patients who will have a good evolution and will be able to leave the care area quickly, especially at times of pressure on the health system.

## 5. Conclusions

Therefore, we conclude, first of all, that in our sample the elevation of hsTnT at the time of IVI diagnosis in the emergency department is significantly associated with increased mortality during the care process.

Secondly, in our sample, the proposed cut-off point of 24 ng/L would allow the safe discharge of patients with IVI confirmed by immunochromatographic analysis and/or genomic amplification techniques, helping to reduce the therapeutic effort and stress on the system.

Thirdly, the retrospective application of our model suggests important economic savings for the health system in the direct cost of care, due to the high percentage of admissions avoided (60.5%), while allowing safe discharge planning from the emergency service for patients with IVI.

hsTnT could be one more block on the retaining wall that we will be forced to develop against viral infections. Paraphrasing Miguel de Cervantes, we can say that in this unpredictable time in which we are living, “being prepared will be half a victory”.

## Figures and Tables

**Figure 1 jpm-12-00520-f001:**
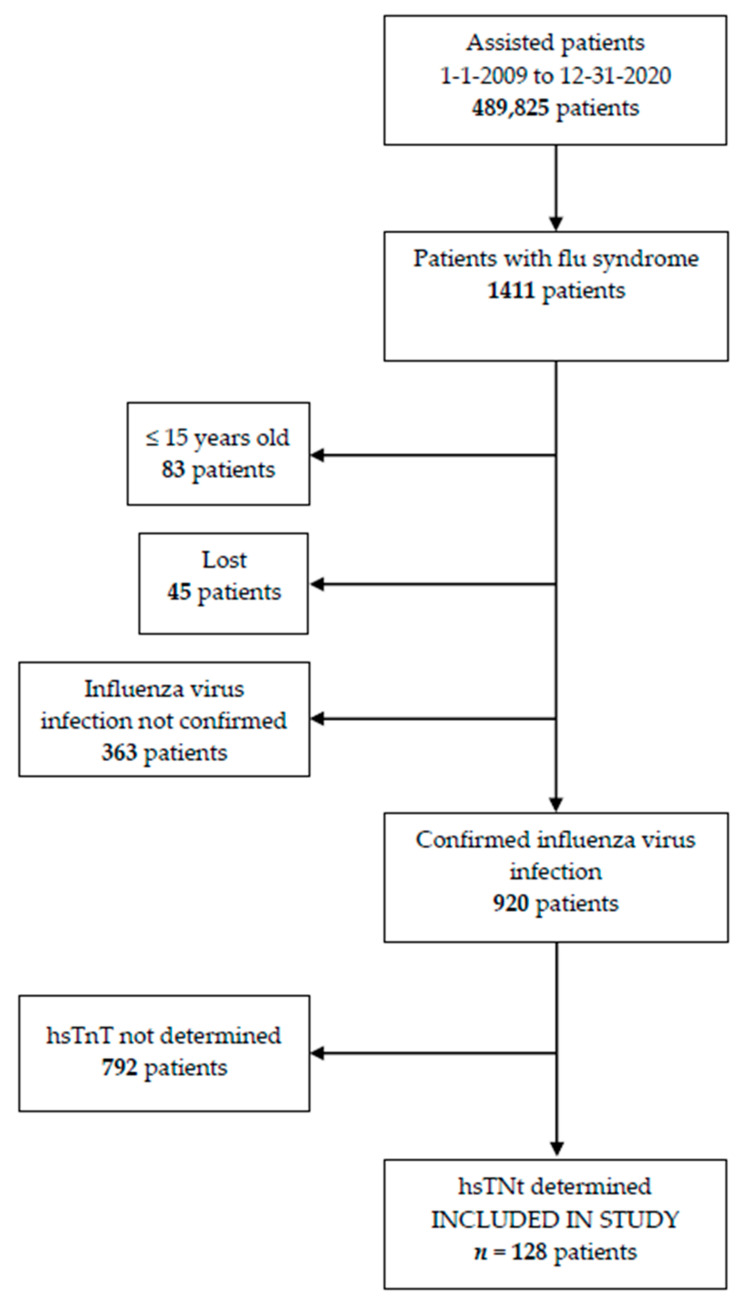
Patients included in the study.

**Figure 2 jpm-12-00520-f002:**
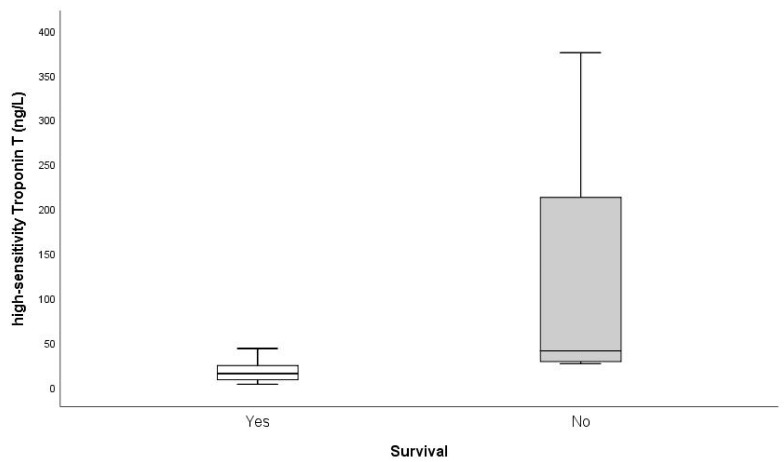
Box plot showing high-sensitivity troponin T values (ng/L) of survivors (*n* = 124) and nonsurvivors (*n* = 4) due to influenza virus infection confirmed by rapid immunochromatographic diagnosis and/or molecular biological diagnosis by techniques. of genomic amplification by polymerase chain reaction methods. Data are presented as medians with 25th and 75th percentiles (boxes) and 95th and 5th percentiles (whiskers).

**Figure 3 jpm-12-00520-f003:**
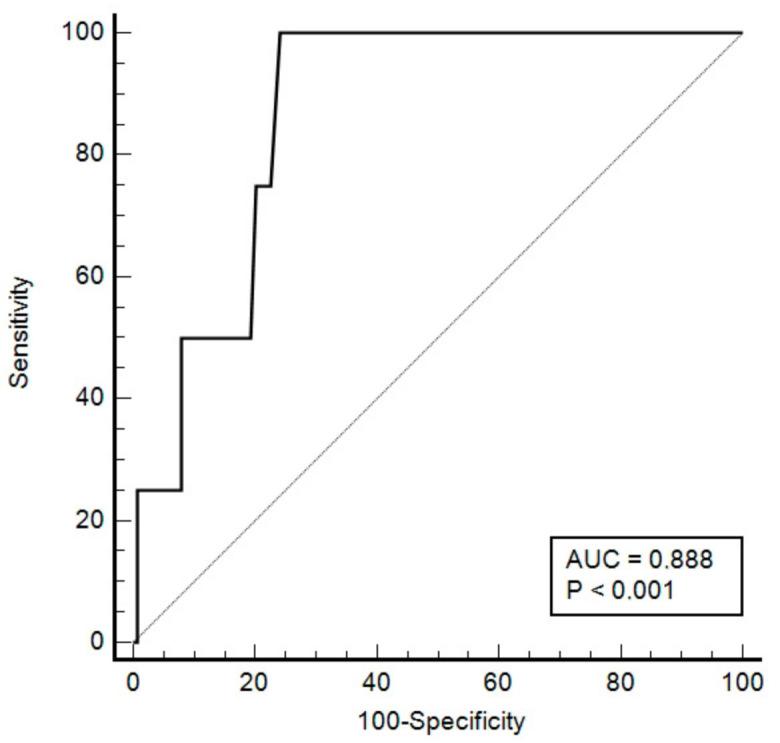
ROC curve plot for ultrasensitive troponin T as a function of mortality. AUC: area under the curve. ROC: receiver operating characteristic.

**Table 1 jpm-12-00520-t001:** General characteristics of the sample.

Chart. General Characteristics of the Sample.					
	SURVIVORS (*n* = 124)		DECEASED (*n* = 4)		*p*-Value
	*n*	%	*n*	%	
**SOCIODEMOGRAPHIC VARIABLES**					
Age (years) [mean (SD)]	68.2	15.7	85.8	4.9	0.008
Female sex	46	37.1	3	75	0.156
**ASSOCIATED COMORBIDITY**					
Active smoker (*n* = 19)	19	15.3	0	0	0.822
Essential hypertension (*n* = 79)	76	61.3	3	75	0.504
DM-1 (*n* = 2)	2	1.6	0	0	0.936
DM-2 (*n* = 28)	28	22.6	0	0	0.546
Dyslipidemia (*n* = 57)	56	45.2	1	25	0.396
Heart failure (*n* = 15)	14	11.3	1	25	0.396
Ischemic heart disease (*n* = 25)	25	20.2	0	0	0.414
Cardiac arrhythmia (*n* = 22)	21	16.9	1	25	0.534
Asthma (*n* = 14)	12	9.7	2	50	0.059
COPD (*n* = 26)	25	20.2	1	25	0.813
Chronic Kidney Disease (*n* = 12)	11	8.9	1	25	0.329
Chronic liver disease (*n* = 12)	11	8.9	1	25	0.546
Cognitive dysfunction (*n* = 7)	5	4	2	50	0.015
Neoplasia (*n* = 11)	10	8.1	1	25	0.305
**CLINICAL PRESENTING VARIABLES**					
Dyspneic feeling (*n* = 65)	61	49.2	4	100	0.062
Body temperature (ºC) [mean (SD)] (*n* = 125)	37.1	0.9	36.8	0.4	0.487
HR (bpm) [median (IQR)] (*n* = 124)	88	28.3	71	7.3	0.035
RF (rpm) [median (IQR)] (*n* = 67)	15	6	14.5	2.5	0.716
TAS (mmHg) [median (IQR)] (*n* = 126)	136	37.3	140	37.3	0.765
DBP (mmHg) [median (IQR)] (*n* = 126)	75	17.3	78.5	22.3	0.440
Pulse oximetry (*n* = 119)	96	4	86.5	4	0.002
**ANALYTICAL VARIABLES**					
Leukocytes x103/µL [median (IQR)] (*n* = 127)	8000	4100	7250	4575	0.310
Neutrophils x103/µL [median (IQR)] (*n* = 125)	5450	3825	5400	1800	0.904
Lymphocytes x103/µL [median (IQR)] (*n* = 125)	900	900	800	700	0.784
Hematocrit % [median (IQR)] (*n* = 127)	41	6.2	41.3	11.3	0.836
Hemoglobin g/dL [median (IQR)] (*n* = 127)	13.6	1.9	13.7	4	0.820
Platelets x103/µL [median (IQR)] (*n* = 126)	176	74	140	99	0.300
Glucose mg/dL [median (IQR)] (*n* = 127)	126	73	144.5	64	0.945
Urea mg/dL [median (IQR)] (*n* = 127)	41	26	38.5	17.8	0.709
Creatinine mg/dL [median (IQR)] (*n* = 126)	0.91	0.33	1.17	0.78	0.337
Sodium mEq/L [median (IQR)] (*n* = 124)	136	4	139	3	0.214
Potassium mEq/L [median (IQR)] (*n* = 120)	4.2	0.7	4.2	0.8	0.372
Bilirubin mg/dL [median (IQR)] (*n* = 56)	0.4	0.4	0.5	0.2	0.941
Prothrombin time % [median (IQR)] (*n* = 119)	88	30	64	75	0.272
Arterial pH [median (IQR)] (*n* = 100)	7.45	0.06	7.41	0.07	0.085
pO2 mmHg [median (IQR)] (*n* = 99)	62	16	53.5	8.5	0.052
pCO2 mmHg [median (IQR)] (*n* = 100)	36.5	9	40.5	8	0.149
**BIOMARKERS**					
CRP mg/dL [median (IQR)] (n = 124)	3.8	8.4	1.9	1.1	0.095
Lactate mg/dL [median (IQR)] (*n* = 22)	14	10	13	0	0.909
NT-proBNP ng/L [median (IQR)] (*n* = 71)	575	2186	3481	4408	0.077
hsTnT ng/L [median (IQR)] (*n* = 128)	14.5	16	40.5	266.5	0.012
**RADIOLOGICAL CHARACTERISTICS**					
Chest X-ray performed (*n* = 124)	120	96.8	4	100	0.936
Parenchymal condensation/infiltrate (*n* = 27)	27	21.8	0	0	0.383
Pleural effusion (*n* = 3)	2	1.6	1	25	0.009
**EVOLUTION VARIABLES**					
Entrance to conventional ward (*n* = 88)	84	67.7	4	100	0.391
Admission to intensive care unit (*n* = 2)	2	1.6	0	0	0.936
Days of survival in deceased [mean (SD)]	-	-	7.3	5.6	-

bpm: beats per minute; brpm: breaths per minute; COPD: chronic obstructive pulmonary disease; CRP: C-reactive protein; DBP: diastolic blood pressure; DM: diabetes mellitus; hsTnT: high-sensitivity troponin T; IQR: interquartile range; NT-proBNP: Amino-terminal fragment of brain natriuretic peptide; Rx: X-rays; SD: standard deviation; TAS: systolic blood pressure.

## Data Availability

The data presented in this study are available on request from the corresponding author.

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
