# Peer review of "High-Sensitivity Troponin T: A Potential Safety Predictive Biomarker for Discharge from the Emergency Department of Patients with Confirmed Influenza"

_jpm, 2022, doi:10.3390/jpm12040520_

Round 1
Reviewer 1 Report
Tazón-Varela MA et al. performed an analytical retrospective observational study to analyze the relationship between the HS-troponin T levels and severity in patients with the influenza virus infection. Authors concluded that “HS-troponin T levels are a good predictor of mortality, and it is useful for safely discharging patients from the emergency department”.
In my viewpoint, this manuscript provides essential information that could be useful for searching biomarkers for the diagnosis and prognosis of TB. However, the current manuscript has major and minor suggestions. Therefore, I suggest changes indicated below and an expert in the English language improve it.
Major comments
- Material and methods section: It Is indicated that enrolled patients’ data were >16 years with usTnT determinate in the emergency room. However, due to the aim of your study, you should include the exact time (hours, days) when the usTnT was measured after they were accepted in the hospital. And if it is possible, I think it also is very important to know the time (days?) when patients started to be ill to when they arrived at the hospital.
- Material and methods section: Did you select patients with the same influenza virus? If it is the same, please specify and if it is not the same, Did you do a sub-analysis to identify if the usTnT levels increased mainly under a specific influenza virus?
- Material and methods section: Indicate the medical reason why usTnT was measured in 128 patients? Did they report a previous cardiac problem? Because 792 did not indicate to measure it. In the first result section, you described that it is because they reported a cardiac problem. So, why did you not use a control group with a high usTnT level but with other influenza-independent respiratory diseases?
The discussion section should include the justification for why you don’t use this group (which is also crucial for interpreting the ROC curve).
- hsTnT (Line 83) means the same that usTnT? Why hereafter did you change the used abbreviation?
Minor comments
- Introduction: Before using an abbreviation, explain what it means (line 73: usTnT; line 74: NPV).
- Homogenise the use of abbreviations, for instance, COVID-19 (Line 57) and Covid-19 (Line 61).
- I suggest dividing the material and methods section using subheads as enrolled patients’ characteristics, diagnostic, statistical analysis and bioethical statement.
- What does NT-proBNP mean? (line 94) …
5. WHO means? (line 239) …
Author Response
ANSWERS TO REVIEWER 1. Major comments.
- Reply to comment 1.
- The hsTnT measurement was performed within the first hour of patient arrival at the emergency department.
- In the material and methods section (line 108), the sentence "hsTnT measurement was performed within the first hour of the patient's arrival at the emergency department" was inserted.
- The time elapsed from the onset of flu-like symptoms to attendance at the emergency department was not recorded.
- The sentence “Nor was the time elapsed from the onset of flu-like symptoms to attendance at the emergency department was not recorded.” was included in the discussion section (line 277).
- Reply to comment 2.
- Viral subtyping was not examined.
- Reply to comment 3
- Because it was a retrospective study and the request for hsTnT analysis was not protocolized in our hospital, the clinicians were free to request the test. However, we refer to this aspect in limitations of the study.
- Reply to comment 4
- This is a typographical error. Actually it is hsTnT.
- Change on line 85. usTnT is replaced by hsTnT.
- This is a typographical error. Actually it is hsTnT.
- The hsTnT measurement was performed within the first hour of patient arrival at the emergency department.
ANSWERS TO REVIEWER 1. Minor comment.
- Reply to comment 1.
- This is a typographical error. Actually it is hsTnT.
- Change on line 85. usTnT is replaced by hsTnT.
- The meaning of the abbreviation NPV, PPV and AUC is introduced.
- Change on line 86. “Negative predictive value” is added.
- Change on line 172. “Positive predictive value” is added.
- Change on line 173. “Area under the ROC curve” is added.
- Reply to comment 2.
- COVID-19 abbreviations are homogenized.
- Change on line 71. Covid-19 is replaced by COVID-19.
- Reply to comment 3.
- The proposed subtitles are introduced in the material and methods section:
- Change on line 89. “Enrolled patients’ characteristics” is added
- Change on line 98. “Diagnostic” is added.
- Change on line 113. “Statistical analysis” is added.
- Change on line 132. “Bioethical statement” is added.
- Reply to comment 4.
- The meaning of NT-proBNP is introduced:
- Change on line 112. “N-terminal prohormone of brain natriuretic peptide” is added.
- Reply to comment 5.
- The meaning of WHO is introduced:
- Change on line 65. “World Health Organization” is added.
- The meaning of WHO is introduced:
- The meaning of NT-proBNP is introduced:
- The proposed subtitles are introduced in the material and methods section:
- COVID-19 abbreviations are homogenized.
- This is a typographical error. Actually it is hsTnT.
Reviewer 2 Report
In this paper, the authors analyzed the relationship between the high-sensitive troponin T (hsTnT) levels in patients with confirmed influenza virus infection (IVI) and mortality during the care process. The authors analysis revealed a link between hsTnT and mortality during the care process. Furthermore, the cut-off point (24 ng/L) obtained in this study was considered to be very useful for the safe discharge of patients.
There are some comments about the content of this paper.
1. At what time point were the samples in this study collected? If the hsTnT value is used as an index of discharge, it may be important to indicate the timing of sample collection.
2. I thought the discussion was verbose. The first half of the discussion may be concise or included in the introduction.
3. I think the decimal point separator in Table 1 is a colon, not a comma.
4. P2. Lane 78: 'usTnT' → 'hsTnT'?
Author Response
ANSWERS TO REVIEWER 2.
- Reply to comment 1.
- The hsTnT measurement was performed within the first hour of patient arrival at the emergency department.
- In the material and methods section (line 108), the sentence "hsTnT measurement was performed within the first hour of the patient's arrival at the emergency department" was inserted.
- Reply to comment 2.
- The introduction and discussion are modified according to the reviewer's instructions.
- In the introduction, line 76 are added: During the last century the healthcare community has observed cardiovascular complications in patients with IVI. The first reference to the increase in cardiovascular morbidity and mortality during IVI epidemics was described in 1932 by Dr. Selwyn D. Collins [10]. Several large epidemiological studies in Russia, the United States, the United Kingdom, and Hong Kong have revealed a temporal association between influenza virus circulation and increased deaths from ischemic heart disease [11]. In 2015 a meta-analysis determined that a recent diagnosis of IVI infection doubled the risk of developing an acute coronary syndrome [12], being able to increase up to 6 times in the seven days after its confirmation [13]. There is increasing evidence that a recent flu infection is associated with an increased risk of developing acute coronary syndrome, heart failure, myocarditis, and arrhythmias, increasing hospital admissions and mortality [12-14]. In a study of 600 patients with confirmed IVI, 86% of the events associated with acute cardiac injury occurred during the three days after flu laboratory ratification [15]. In short, IVI infection can predispose to suffering a coronary event in the following days. IVI can also cause other cardiovascular complications, multiplying 3-5 times the risk of heart failure, acute lung edema and arrhythmias, especially in the first three days of infection [16].
- In the discussion, line 179 are deleted: During the last century the healthcare community has observed cardiovascular complications in patients with IVI. The first reference to the increase in cardiovascular morbidity and mortality during IVI epidemics was described in 1932 by Dr. Selwyn D. Collins [11]. There is increasing evidence that a recent flu infection is associated with an increased risk of developing acute coronary syndrome, heart failure, myocarditis, and arrhythmias, increasing hospital admissions and mortality [12-14]. Several large epidemiological studies in Russia, the United States, the United Kingdom, and Hong Kong have revealed a temporal association between influenza virus circulation and increased deaths from ischemic heart disease [13]. In 2015 a meta-analysis determined that a recent diagnosis of IVI infection doubled the risk of developing an acute coronary syndrome [15], being able to increase up to 6 times in the seven days after its confirmation [12]. In a study of 600 patients with confirmed IVI, 86% of the events associated with acute cardiac injury occurred during the three days after flu laboratory ratification [16]. In short, IVI infection can predispose to suffering a coronary event in the following days. IVI can also cause other cardiovascular complications, multiplying 3-5 times the risk of heart failure, acute lung edema and arrhythmias, especially in the first three days of infection [17].
- The references have been modified to adapt to the bibliography.
- Reply to comment 3.
- It has been modified in table 1.
- Reply to comment 4.
- This is a typographical error. Actually it is hsTnT.
- Change on line 85. usTnT is replaced by hsTnT.
- This is a typographical error. Actually it is hsTnT.
- The introduction and discussion are modified according to the reviewer's instructions.
- The hsTnT measurement was performed within the first hour of patient arrival at the emergency department.

Round 2
Reviewer 1 Report
The authors have satisfactory replied to all my suggestions.